# Multimodal Iterative RAG for Knowledge-Intensive Visual Question Answering

## Abstract

Recent advances in Multimodal Large Language Models (MLLMs) have significantly enhanced the ability of these models in multimodal understanding and reasoning. However, the performance of MLLMs for knowledge-intensive visual questions, which require external knowledge beyond the visual content of an image, still remains limited. While Retrieval-Augmented Generation (RAG) has become a promising solution to provide models with external knowledge, its conventional single-pass framework often fails to gather sufficient knowledge. To overcome this limitation, we propose MI-RAG, a Multimodal Iterative RAG framework that leverages reasoning to enhance retrieval and incorporates knowledge synthesis to refine its understanding. At each iteration, the model formulates a reasoning-guided multi-query to explore multiple facets of knowledge. Subsequently, these queries drive a joint search across heterogeneous knowledge bases, retrieving diverse knowledge. This retrieved knowledge is then synthesized to enrich the reasoning record, progressively deepening the model's understanding. Experiments on challenging benchmarks, including Encyclopedic VQA, InfoSeek, and OK-VQA, show that MI-RAG significantly improves both retrieval recall and answer accuracy, establishing a scalable approach for compositional reasoning in knowledge-intensive VQA.

## 1 Introduction

The emergence of Multimodal Large Language Models has driven significant progress in multimodal understanding and reasoning. Nonetheless, the performance of MLLMs remains constrained in knowledge-intensive Visual Question Answering (VQA) tasks, which require external knowledge beyond the visual content in the image. Answering such questions is particularly challenging as it involves: (1) grounding visual entities in the image to the textual question, (2) collecting text knowledge relevant to both the entities and the question, and (3) synthesizing the visual entities and corresponding textual knowledge to derive the answer.

To address this challenge, multimodal Retrieval-Augmented Generation has emerged as a promising solution. The conventional *retrieve-then-read* process starts with a retriever using the input image and question to find relevant image-text pairs from a knowledge base (KB). A reader model then synthesizes this information to create the final answer. Research on optimizing this single-step process has largely advanced along two fronts: improving initial retrieval with high-quality embeddings Lin et al. (2024); Wei et al. (2024); Jiang et al. (2025); Lin et al. (2025); Jiang et al. (2024b); Zhang et al. (2024b), and refining retrieval results by selection and reranking Cocchi et al. (2025); Zhang et al. (2024a); Ling et al. (2025); Chen et al. (2024); Liu et al. (2024b); Yan & Xie (2024); Yang et al. (2025).

Although prior work has achieved notable improvements within the *retrieve-then-read* process, the process itself has inherent limitations for knowledge-intensive VQA. A single retrieval step with an initial query is often insufficient, as imperfect retrievers may fail to gather all necessary knowledge at once Shi et al. (2023); Zhang et al. (2023); Cuconasu et al. (2024). Furthermore, a single reasoning step to ground visual entities and synthesize knowledge is problematic, as distracting or irrelevant content can mislead synthesis and influence the final answer Petroni et al. (2020); Wu et al. (2024); Amiraz et al. (2025); Huang et al. (2025); Wu et al. (2025).

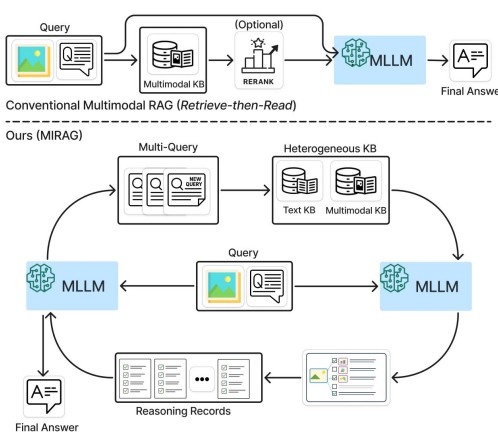

Figure 1: An overview of MI-RAG. Unlike the *retrieve-then-read* process of conventional multimodal RAG, MI-RAG iteratively refines its reasoning and retrieval using a multi-query search over heterogeneous KBs.

Motivated by these limitations, the text domain has shifted to *iterative RAG*, a paradigm that leverages an LLM's reasoning through multiple rounds of retrieval and reasoning. In this iterative cycle, the model leverages its reasoning to guide intermediate steps, which transform queries to collect more relevant knowledge and synthesize newly retrieved knowledge to progressively update the reasoning itself. Shao et al. (2023); Trivedi et al. (2022); Xiong et al. (2024); Liu et al. (2024a); Yue et al. (2025); Gao et al. (2025). While previous work has shown promise with this approach in the text-only domain, it has largely focused on refining a single subquery of decomposing multi-hop questions at each step. Extending this to the more complex multimodal domain remains an open challenge.

This challenge arises because a naive adaptation of the *iterative RAG* is insufficient for the multimodal domain, which introduces two distinct considerations: (i) it requires retrieving diverse knowledge by exploring multiple facets of a visually grounded entity and its related textual knowledge across modalities, and (ii) it demands compositional reasoning to synthesize the visual entity with its corresponding knowledge by integrating a diverse set of visual-to-visual, visual-to-text, and text-to-text factual links. This highlights the need for a more specialized approach that can effectively handle the complexities of both visual and textual knowledge.

In response to these multimodal domain considerations, we propose Multimodal Iterative Retrieval-Augmented Generation (MI-RAG). As illustrated in Figure 1, MI-RAG is an iterative framework that cyclically refines its query and reasoning by alternately performing reasoning-guided query transformation and retrieval-augmented reasoning. Our framework makes two primary contributions. First, we employ a reasoning-guided multi-query that dynamically searches multiple facets across modalities. Second, these queries drive a joint search across heterogeneous KBs to find a diverse set of factual links, allowing MI-RAG to compose visually-grounded knowledge and related textual knowledge. Through the synergy of multi-query and heterogeneous KBs, MI-RAG is able to retrieve sufficient knowledge and enable robust compositional reasoning over diverse knowledge across modalities. Through iterations, MI-RAG collects diverse knowledge progressively with accumulating a reasoning record, where each record is the result of compositional reasoning over newly retrieved knowledge, thereby refining the model's understanding to guide the subsequent iteration step.

## 2 RELATED WORK

### 2.1 MULTIMODAL RAG FOR KNOWLEDGE-INTENSIVE VQA

The dominant paradigm for multimodal RAG follows the *retrieve-then-read* process. Initial work focused on enhancing the retrieval phase, primarily by learning more effective multimodal embeddings. Pioneering approaches aligned queries and text documents using contrastive learning Luo et al. (2023), while others fused similarity scores from pretrained CLIP Wei et al. (2024). To improve retrieval, subsequent methods employed fine-grained late interaction between visual and textual tokens Lin et al. (2023; 2024) or leveraged MLLMs with instruction tuning to produce multimodal embeddings Jiang et al. (2025); Liu et al. (2024b); Zhang et al. (2024b). A separate line of research has explored generative retrieval, using document identifiers Long et al. (2024) or enhancing generative retrieval via reinforcement learning Long et al. (2025).

Another line of research improves the read phase, typically by reranking or refining the retrieved candidates. Hierarchical systems first perform a coarse, image-based search and then refine the results with text-based or multimodal rerankers Caffagni et al. (2024); Yan & Xie (2024); Yang

et al. (2025). Others leverage the reasoning capabilities of MLLMs to self-reflect on retrieval needs and evaluate relevance Cocchi et al. (2025), generate semantic tags for consistency and reranking based on relevance scores Ling et al. (2025). While effective, these methods largely adhere to the *retrieve-then-read* process. In contrast, MI-RAG introduces a dynamic loop that iterates between reasoning-guided query refinement and retrieval-augmented reasoning to progressively solve knowledge-intensive VQA tasks.

## 2.2 ITERATIVE RAG

When single-step retrieval is insufficient for complex, multi-hop questions, *iterative RAG* has emerged as a powerful paradigm in the text-only domain. These methods typically use an iterative loop to refine queries and retrieve new knowledge to improve subsequent iterations. Query transformation techniques are central to this process and include query expansion, which uses pseudo-answers to formulate richer queries Shao et al. (2023); Wang et al. (2024); query generation, which decomposes complex questions into simpler, answerable sub-queries Zhang et al. (2025); Liu et al. (2024a); and query planning, where a model strategically decides when and what to retrieve Trivedi et al. (2022); Yu et al. (2024).

To guide the subsequent query transformation, the LLM performs retrieval-augmented reasoning. This involves synthesizing newly retrieved information into a reasoning record. Common strategies include summarizing retrieved passages Jiang et al. (2024c); Zhang et al. (2025), extracting direct evidence snippets Yu et al. (2024), or generating pseudo-answers for intermediate sub-queries, often with chain-of-thought prompting Shao et al. (2023); Trivedi et al. (2022). However, these text-based iterative methods are primarily designed for multi-hop question decomposition, where the goal is to build a linear reasoning chain. Our work addresses knowledge-intensive VQA, which requires the composition of diverse knowledge across modalities.

## 3 METHOD

### 3.1 FRAMEWORK OF MULTIMODAL ITERATIVE RAG

---

**Algorithm 1** MI-RAG

**Require:** $I, Q, \text{KB}_\text{T}, \text{KB}_\text{M}, \text{MLLM}, N$
**Ensure:** Final answer $A$

1: $D \leftarrow \text{MLLM}(Q, I)$
2: $Q_0^E \leftarrow \text{Expand}(Q, D)$
3: $K_\text{T} \leftarrow \text{Ret}(\text{KB}_\text{T}, Q_0^E)$
4: $K_\text{M} \leftarrow \text{Ret}(\text{KB}_\text{M}, I, Q_0^E)$
5: $R_0 \leftarrow \text{Reasoning}(\text{MLLM}, Q, I, K_\text{T}, K_\text{M}, D)$
6: **for** $i = 1$ **to** $N$ **do**
7: $\quad Q_i^E \leftarrow \text{Expand}(Q, R_{i-1})$
8: $\quad Q_i^G \leftarrow \text{Generate}(\text{MLLM}, Q, R_0, \ldots, R_{i-1})$
9: $\quad K_\text{T} \leftarrow \text{Ret}(\text{KB}_\text{T}, Q_i^E)$
10: $\quad K_\text{T} \leftarrow K_\text{T} \cup \text{Ret}(\text{KB}_\text{T}, Q_i^G)$
11: $\quad K_\text{M} \leftarrow \text{Ret}(\text{KB}_\text{M}, I, Q_i^E)$
12: $\quad K_\text{M} \leftarrow K_\text{M} \cup \text{Ret}(\text{KB}_\text{M}, I, Q_i^G)$
13: $\quad R_i \leftarrow \text{Reasoning}(\text{MLLM}, Q, I, K_\text{text}, K_\text{mm})$
14: **end for**

15: $A \leftarrow \text{MLLM}(Q, I, \{R_0, R_1, \ldots, R_N\})$
16: **return** $A$

---

As detailed in Algorithm 1, the MI-RAG framework processes multimodal queries (an image I and question Q), through iterative retrieval and reasoning. For conciseness in the pseudocode, we use the following abbreviations: $\text{KB}_\text{T}$ for text KB, $\text{KB}_\text{M}$ for multimodal KB, $K_\text{T}$ for text passages, $K_\text{M}$ for image-text pairs, $\text{Ret}$ for Retrieve, $D$ for image description, $Q^E$ for expanded query, $Q^G$ for generated sub-query, $R$ for reasoning record and $N$ for iteration counts. This iterative process begins with the generation of initial reasoning records (Lines 1–5; Section 3.2). Subsequently, in each iteration, MI-RAG leverages the accumulated reasoning record to formulate the multi-query (Lines 7–8; Section 3.3) and perform a joint search using the multi-query across heterogeneous KBs (Lines 9–12; Section 3.4), thereby dynamically gathering diverse and relevant knowledge. Following retrieval, MI-RAG updates the reasoning record by summarizing newly acquired knowledge about the queries (Line 13; Section 3.5). Finally, the model derives the final answer over the accumulated records (Line 15; Section 3.6).

### 3.2 INITIAL REASONING RECORD GENERATION

The initial reasoning record is generated by leveraging the retrieved knowledge with internal MLLM knowledge. Specifically, given a multimodal query, the MLLM generates an image description $D$

corresponding to the question. We perform query expansion, yielding expanded query $Q_0^E = [Q, D]$. This query $Q_0^E$ is then used to retrieve knowledge from our heterogeneous KBs, collecting both text passages and image-text pairs. Finally, the model synthesizes retrieved diverse knowledge and descriptions into a reasoning record. This initial process generates the first reasoning record to bootstrap our iterative framework.

### 3.3 Reasoning-guided Multi-query Transformation

MI-RAG employs a reasoning-guided multi-query transformation that leverages the MLLM's intermediate reasoning steps to dynamically reformulate the query. This process uses the accumulated reasoning records to dynamically generate a set of complementary queries, ensuring a comprehensive search for multifaceted knowledge across modalities:

**Query Expansion**: This method leverages the latest reasoning record to expand the original query. The resulting query is designed to reflect the model's latest reasoning, which enhances retrieval performance while preserving the original question's intent.

**Query Generation**: This method produces a set of sub-queries designed to explore alternative reasoning paths, resolve ambiguities, or explicitly seek out missing facts identified during the analysis of the reasoning record.

We provide details of the prompt templates used to guide this transformation in Appendix C.

### 3.4 Joint Search Across Heterogeneous KBs

To enhance compositional reasoning, it is crucial to retrieve diverse knowledge across modalities. To acquire the a diverse set of factual knowledge, we perform a joint search over two heterogeneous KBs: a multimodal KB and a textual KB. The multimodal KB provides image-text pairs for visual grounding of entities, while the textual KB offers broad knowledge that extends beyond these pairs. Each query from the multi-query is used to retrieve candidates from both KBs simultaneously. To balance the token budget with broad knowledge coverage, our strategy retrieves twice as many text passages as image-text pairs, collecting up to 20 passages and 10 image-text pairs per iteration. The implementation details for multimodal retrieval are provided in Appendix D.

**Retrieval from textual KB**: The textual KB is queried using the multi-query. A text embedding model is used to retrieve the top-k passages based on semantic similarity.

**Retrieval from multimodal KB**: In contrast to prior methods that often rely on large, specialized retrievers, MI-RAG employs a lightweight retrieval strategy. We utilize a multimodal similarity scoring approach Changin et al. (2025) to support query transformation. For each candidate image-text pair, the similarity score is the average of two components: (1) a text-to-text similarity between the candidate's text and the transformed query; and (2) a fixed image-to-image similarity between the candidate's image and the input query image, which maintains constant visual relevance across iterations.

### 3.5 Generate Reasoning Record with Knowledge

At each iteration, a reasoning record is generated by summarizing the newly retrieved knowledge. The MLLM synthesizes relevant evidence from visual entities and related text knowledge. This process utilizes the MLLM's compositional reasoning to generate a reasoning record that captures relevant evidence. By iteratively enriching its understanding with these reasoning records, the model maintains a coherent reasoning path and enables more precise query transformations in subsequent steps. We provide details of the prompt in Appendix C.

### 3.6 Final Answer Generation

Upon completion of the $N$ iterative cycles, MI-RAG generates the final answer. In this stage, MLLM synthesizes the original question $Q$, the input image $I$, and the accumulated reasoning records $\{R_0, \ldots, R_N\}$ to produce an evidence-based answer.

## 4 EXPERIMENTS

### 4.1 EXPERIMENT SETUP

**Datasets:** Our evaluation is conducted on three diverse knowledge-intensive VQA benchmarks: validation split of InfoSeek Chen et al. (2023), validation split of OK-VQA Marino et al. (2019), and test split of Encyclopedic VQA Mensink et al. (2023). For InfoSeek, we follow the setup of PreFLMR Lin et al. (2024) and utilize the validation split provided by the M2KR benchmark.

**Knowledge Bases:** To provide the necessary external knowledge, our framework retrieves text passages and image-text pairs from heterogeneous KBs, which are held constant across all experiments: a Wikipedia text corpus provided by FlashRAG Jin et al. (2024) for text KB and a 2M image-text pairs from Encyclopedic VQA for multimodal KB.

**Models:** To assess how performance scales with reasoning ability, we primarily use two model variants: Gemma-3 Team et al. (2025) and Gemini-2.5-Flash Comanici et al. (2025). Our framework employs lightweight retrievers. For the multimodal retrieval of image-text pairs, we compute a multimodal similarity score using image embeddings from SigLIP2-SO-400m Tschannen et al. (2025) and text embeddings from ModernBERT Warner et al. (2024). For the retrieval of text passages, we use E5-base-v2 Wang et al. (2022), following the setup in FlashRAG Jin et al. (2024).

**Evaluation Metrics:** We evaluate MI-RAG using standard metrics for each benchmark to evaluate accuracy and retrieval recall. For accuracy, we report the BERT matching score (BEM) Bulian et al. (2022) on Encyclopedic VQA. On the InfoSeek dataset, we measure Exact Match (EM) accuracy. To constrain the model's answer to a single exact entity for this metric, we employ a 4-shot prompting strategy using examples from the training split. The full prompt template and further details are available in Appendix E. For OK-VQA, we use the standard VQA-score Marino et al. (2019) and a Cover Exact Match (Cover EM) Jiang et al. (2024a); Yue et al. (2025), which checks if the ground-truth answer is contained within the response. To assess retrieval performance, we use two metrics based on ground-truth annotation availability. Retrieval quality is measured using Recall of ground-truth entities where available, and Pseudo-Relevance Recall (PRR) otherwise Luo et al. (2021).

### 4.2 RETRIEVAL PERFORMANCE ON VQA BENCHMARKS

Table 1: Recall comparison on the InfoSeek and Encyclopedic VQA. The FT column with ✓ indicates methods fine-tuned on the target dataset. Methods marked with † employ a reranking step after retrieval. Our vanilla method (‡) uses a single query with expansion to retrieve $K$ pairs from the multimodal KB, reporting average recall across iterations for fair comparison. Our full framework (*) reports the cumulative recall after 4 iterations. We report it on R@5, as each query retrieves fewer than five pairs on average. The best and second-best overall accuracies are highlighted in **bold** and with an underline, respectively.

| FT | Method | InfoSeek Validation | | | Encyclopedic VQA | | |
|---|---|---|---|---|---|---|---|
| | | R@5 | R@10 | R@20 | R@5 | R@10 | R@20 |
| ✗ | CLIP ViT-L/14 Radford et al. (2021) | 54.0 | 61.6 | 68.6 | 7.7 | 12.1 | 16.5 |
| ✗ | SigLIP2-So400m Tschannen et al. (2025) | 52.5 | 60.2 | 68.3 | 30.8 | 36.6 | 41.9 |
| ✗ | EVA-CLIP-8B Sun et al. (2023) | 67.1 | 73.0 | 77.9 | 31.3 | 41.0 | 48.8 |
| ✓ | Wiki-LLaVA Caffagni et al. (2024) | - | 66.1 | 71.9 | - | 9.9 | 13.2 |
| ✓ | LLM-RA Jian et al. (2024) | 53.8 | - | - | - | - | - |
| ✓ | mR²AG† Zhang et al. (2024a) | - | 65.0 | 71.0 | - | - | - |
| ✓ | ReflectiVA† Cocchi et al. (2025) | 77.6 | - | 86.4 | 36.1 | - | 49.8 |
| ✓ | EchoSight† Yan & Xie (2024) | 74.0 | 77.4 | 77.9 | 47.9 | 48.8 | 48.8 |
| ✓ | OMGM Yang et al. (2025) | 73.9 | 80.0 | 84.8 | 41.2 | 49.8 | **58.7** |
| ✓ | OMGM† Yang et al. (2025) | **80.8** | 83.6 | 84.8 | **55.7** | **58.1** | **58.7** |
| ✗ | ReAuSE Long et al. (2025) | 59.5 | - | - | - | - | - |
| ✗ | Ours (Gemma-3-4B)‡ | **80.8** | **85.6** | **89.7** | 39.1 | 44.3 | 49.8 |
| ✗ | Ours (Gemma-3-4B)* | 88.0 | - | - | 53.6 | - | - |
| ✗ | Ours (Gemini-Flash-2.5)* | **92.1** | - | - | **62.4** | - | - |

As shown in Table 1, our MI-RAG framework demonstrates effective retrieval without any task-specific fine-tuning. We analyze its performance by highlighting three key findings: the effectiveness of reasoning-guided queries in a lightweight setup, MI-RAG's performance boost from the iterative process, and the framework's inherent synergy with larger MLLMs.

To evaluate our reasoning-guided query transformation, we provide a vanilla method (‡), which uses a single query with a multimodal KB, a small 4B MLLM, and a lightweight retriever for fair comparison. This contrasts with approaches that employ larger retrievers and use MLLM for reranking. Even under these constraints, our method is highly effective, achieving a top R@20 score of 89.7% on InfoSeek and the second-highest R@20 on Encyclopedic VQA.

Our full framework (*) performs multiple rounds of multi-query searches across heterogeneous KBs to progressively accumulate multifaceted knowledge across modalities. The results show significant gains in cumulative recall, reaching 88.0% (+7.2%) on InfoSeek and 53.6% (+14.5%) on Encyclopedic VQA.

Finally, we analyze MI-RAG's scalability by evaluating its two core components: the MLLM and the retriever. As shown in Table 1, scaling the MLLM from Gemma-3-4B to Gemini-Flash-2.5 consistently improves performance. This result confirms that a more capable MLLM enhances both query transformation and compositional reasoning. In Section 6.1, we demonstrate that scaling the retriever provides a further performance boost.

Table 2: Recall comparison on the OK-VQA benchmark when utilizing Wikipedia as the knowledge source. Symbols denote: fine-tuning (✓); reranking (†); our vanilla baseline (‡); and our MI-RAG framework (*). The best and second-best overall accuracies are highlighted in **bold** and with an underline, respectively.

| FT | Methods | OK-VQA | | |
|---|---|---|---|---|
| | | PRR@5 | PRR@10 | PRR@20 |
| ✓ | ReViz-ICT | 61.9 | 72.6 | 81.1 |
| ✓ | GeMKR | 70.8 | 79.1 | - |
| ✓ | FLMR | 68.1 | 78.0 | - |
| ✓ | Pre-FLMR | 68.6 | - | - |
| ✓ | ReAuSE | **88.0** | **91.3** | - |
| × | OMGM† | 73.4 | - | - |
| × | Ours (Gemma-3-4B)‡ | 77.1 | 85.4 | **91.2** |
| × | Ours (Gemma-3-4B)* | 95.2 | - | - |
| × | Ours (Gemini-Flash-2.5)* | **97.1** | - | - |

On the OK-VQA benchmark, MI-RAG demonstrates strong retrieval performance, as detailed in Table 2. Our vanilla method (‡), despite its lightweight setup, shows competitive performance. Enabling the full MI-RAG provides a substantial boost to 95.2% (18.1%) for the same 4B model. A more powerful MLLM with stronger reasoning achieves 97.1% recall, outperforming all prior work.

This robust performance contrasts sharply with prior methods, which often exhibit a performance trade-off across different benchmarks. For instance, ReAuSE excels in its trained knowledge domain (OK-VQA) but fails to generalize to InfoSeek, whereas OMGM shows the reverse pattern. In contrast, our method shows high performance across all benchmarks.

### 4.3 Accuracy on VQA Benchmarks

As demonstrated in Table 3, our MI-RAG framework establishes a new state-of-the-art on the Encyclopedic VQA benchmark. To isolate the contribution of our iterative reasoning, we first evaluate a setting with a strong initial retrieval provided by Google Lens. This configuration reveals that performance scales directly with the MLLM's reasoning capacity: accuracy improves from a competitive 62.8% with Gemma-3-4B to a state-of-the-art 75.5% with Gemini-2.5-Flash. This highlights the synergy between a high-quality initial retrieval and the model's ability to iteratively refine it, an analysis further explored in Section 6.1.

In the more challenging setting without Google Lens, we use reasoning records generated by Gemini-2.5-Flash with our lightweight retriever setup. When these records are applied to LLaVA-MORE-8B checkpoint from ReflectiVA as the answer, our framework achieves a competitive accuracy of 50.4%. When Gemini-2.5-Flash is used as the answerer, our method achieves 61.4% accuracy.

On the InfoSeek benchmark, MI-RAG translates its strong retrieval performance into state-of-the-art accuracy, as detailed in Table 4. Our method achieves 48.0% accuracy with Gemini-2.5-Flash and improves to 51.7% with GPT-4o as the answerer, outperforming all prior work. This is driven by our iterative process, where accumulated reasoning records enable robust compositional reasoning over

Table 3: Accuracy comparison on the Encyclopedic VQA benchmark. A checkmark (✓) indicates fine-tuning, a dagger (†) denotes MLLM-based reranking, and an asterisk (*) marks the use of ReflectiVA checkpoints. Our framework's Gemma-series models generate their own reasoning records, whereas other models use records provided by Gemini-2.5-Flash.

| Google Lens | FT | Method | Model | Single-hop | All |
|---|---|---|---|---|---|
| | ✓ | Wiki-LLaVA Caffagni et al. (2024) | LLaVA-v1.5-7B | 21.8 | – |
| | ✓ | EchoSight† Yan & Xie (2024) | Mistral-7B | 35.5 | – |
| | ✓ | ReflectiVA† Cocchi et al. (2025) | LLaVA-MORE-8B | 35.5 | – |
| | ✓ | MMKB-RAG† Ling et al. (2025) | Qwen2-VL-7B | 39.7 | 35.9 |
| ✗ | ✓ | OMGM† Yang et al. (2025) | LLaVA-v1.5-7B | 50.2 | - |
| | ✓ | OMGM† Yang et al. (2025) | GPT-4o | 51.2 | - |
| | ✓ | Ours | LLaVA-MORE-8B* | 50.4 | 49.2 |
| | ✗ | Ours | Gemma-3-4B | 48.4 | 52.4 |
| | ✗ | Ours | Gemini-2.5-Flash | **61.4** | **62.9** |
| | ✗ | HAMMR Castrejon et al. (2024) | PaLI-X | 47.8 | – |
| | ✗ | LVLM cascade Alazraki et al. (2023) | PaLM, GPT-3, PaLI | 53.4 | – |
| ✓ | ✓ | mR²AG† Zhang et al. (2024a) | LLaVA-v1.5-7B | 55.9 | 51.8 |
| | ✗ | Ours | Gemma-3-4B | 62.8 | 63.7 |
| | ✗ | Ours | Gemini-2.5-Flash | **75.5** | **70.8** |

Table 4: Accuracy comparison on the InfoSeek benchmark: the top section evaluates the full validation set, while the bottom section assesses a 5K validation split. A checkmark (✓) in the FT column indicates fine-tuning, a dagger (†) denotes a reranking step, and an asterisk (*) marks the use of checkpoints from ReflectiVA. Our framework's Gemma-series models generate their own reasoning records, whereas other models use records provided by Gemini-2.5-Flash.

| FT | Method | Retriever | Model | Acc |
|---|---|---|---|---|
| ✓ | Wiki-LLaVA Caffagni et al. (2024) | CLIP-ViT-L | LLaVA-1.5-7B | 28.9 |
| ✓ | EchoSight† Yan & Xie (2024) | EVA-CLIP-8B | LLaMA3-8B | 31.3 |
| ✓ | mR²AG† Zhang et al. (2024a) | CLIP-ViT-L | LLaVA-1.5-7B | 40.2 |
| ✓ | ReflectiVA† Cocchi et al. (2025) | EVA-CLIP-8B | LLaVA-MORE-8B | 40.1 |
| ✓ | MMKB-RAG† Ling et al. (2025) | PreFLMR ViT-G | Qwen2-VL-7B | 36.7 |
| ✓ | OMGM† Yang et al. (2025) | EVA-CLIP-8B | LLaVA-1.5-7B | 43.5 |
| ✓ | OMGM† Yang et al. (2025) | EVA-CLIP-8B | GPT-4o | 42.1 |
| ✓ | RA-VQA-v2 Lin et al. (2024) | PreFLMR ViT-G | BLIP2-Flan-T5-XL | 30.7 |
| ✓ | MMKB-RAG Ling et al. (2025) | PreFLMR ViT-G | Qwen2-VL-7B | 34.7 |
| ✓ | MUKA Deng et al. (2025) | MUKA | VILA-13B | 42.5 |
| ✓ | | | LLaVA-MORE-8B* | 45.1 |
| ✗ | Ours | SigLIP2-SO-400m | Gemma-3-4B | 33.1 |
| ✗ | | | Gemini-2.5-Flash | 48.0 |
| ✗ | | | GPT-4o | **51.7** |

multifaceted, diverse knowledge. The strength of this iterative reasoning is further demonstrated by its ability to improve ReflectiVA's LLaVA-MORE-8B checkpoint to 45.1%.

Table 5: Comparison of accuracy based on two metrics: Cover EM, which checks if the ground-truth answer is included in the response, and Exact Match, which requires the response to match the ground-truth answer exactly.

| Model | InfoSeek Validation | | InfoSeek Human | | OK-VQA | |
|---|---|---|---|---|---|---|
| | EM | CEM | EM | CEM | EM | CEM |
| Gemma-3-4B | 33.1 | 50.4 | 32.3 | 39.3 | 43.3 | 68.8 |
| Gemini-2.5-Flash | 48.0 | 61.3 | 40.5 | 44.9 | 57.3 | 71.0 |

Recent MLLMs generate detailed, free-form responses by leveraging multi-stage post-training to better align with human preferences, making strict exact match (EM) an inadequate metric for a comprehensive evaluation, as noted in prior work Shao et al. (2023); Cocchi et al. (2025). Consequently, metrics like Cover EM (CEM), which verify the inclusion of the ground-truth answer within a longer response, are increasingly used Jiang et al. (2024a); Yue et al. (2025); Wang et al. (2025). This gap arises

when MLLM is not explicitly fine-tuned to produce a strict single-entity answer. The MLLM's tendency to generate a rationale before the final answer results in penalties under the EM metric.

# 5  ABLATIONS

We conducted ablation studies to evaluate MI-RAG's core components. For efficiency, all experiments are performed on downsampled subsets of InfoSeek and Encyclopedic VQA using Gemini-2.5-Flash as the MLLM. We strictly controlled the retrieval budget to ensure a fair comparison. For our default heterogeneous KB setting, we retrieve 20 text passages and 10 image-text pairs. To maintain a comparable budget for multimodal KB, we retrieve 20 pairs. This total retrieval budget is held constant across both single-query and multi-query. We report performance using cumulative recall, along with CoverEM on InfoSeek and BEM on Encyclopedic VQA. We provide details of the downsampling algorithm in Appendix G.

## 5.1  ABLATION OF COMPONENTS

Table 6: Ablation of MIRAG on the InfoSeek validation subset and Encyclopedic VQA test subset.

| Components | | | InfoSeek | | Encyclopedic VQA | |
|---|---|---|---|---|---|---|
| Multi Query | Hetero. KB | Iter. | Acc | Recall | Acc | Recall |
| × | × | × | 46.70 | 87.28 | 50.78 | 47.06 |
| ✓ | × | × | 49.44 | 90.34 | 52.94 | 49.02 |
| × | ✓ | × | 49.21 | 88.08 | 51.18 | 52.75 |
| ✓ | ✓ | × | 54.11 | 91.43 | 56.67 | 57.25 |
| ✓ | ✓ | ✓ | **61.84** | **94.36** | **63.33** | **67.25** |

Our ablation analysis in Table 6 reveals the impact of each component. We establish a strong baseline using a vanilla RAG system over a multimodal KB, enhanced with our initial query expansion, which integrates the MLLM's knowledge. Introducing the reasoning-guided multi-query yields a substantial accuracy gain by gathering the multifaceted knowledge. Moreover, employing our heterogeneous KB is crucial for compositional reasoning, as it enables the model to compose visually-grounded knowledge with the broad textual knowledge. Combining these two components creates a crucial synergy for compositional reasoning. However, the iterative process delivers the most significant improvement across all datasets.

## 5.2  IMPACT OF ITERATIVE REFINEMENT

Table 7: Performance over iterations on the InfoSeek and Encyclopedic VQA.

| Iter. | InfoSeek | | Encyclopedic VQA | |
|---|---|---|---|---|
| | Acc | Recall | Acc | Recall |
| 1 | 55.72 | 92.13 | 58.46 | 65.10 |
| 2 | 59.10 | 92.95 | 60.86 | 66.27 |
| 3 | 61.41 | 93.44 | 61.96 | 66.86 |
| 4 | 61.84 | 94.36 | 63.33 | 67.25 |

To further investigate the benefits of iterative refinement, we analyze performance across multiple iterations, with results presented in Table 7. The study confirms that iterative processing is a primary driver of accuracy, and performance consistently improves over the vanilla baseline across all datasets. The most substantial gains are typically observed within the first two iterations, validating the efficacy of the core refinement loop. As visualized in Figure 2, these gains gradually saturate.

## 5.3  IMPACT OF HETEROGENEOUS KB

Table 8: Ablation on the KB configuration, showing performance after 4 iterations. Heterogeneous KB refers to the use of both KBs, while Multimodal restricts retrieval to the multimodal KB only.

| KB Config. | InfoSeek | | Encyclopedic VQA | |
|---|---|---|---|---|
| | Acc | Recall | Acc | Recall |
| Multimodal | 57.49 | 91.79 | 56.86 | 58.63 |
| Heterogenous | 61.84 | 94.36 | 63.33 | 67.25 |

As shown in Table 8, we evaluate the impact of heterogeneous KB. Applying our reasoning-guided multi-query approach to the multimodal KB alone yields high recall, demonstrating its ability to retrieve visually grounded information. However, its precision remains limited, indicating that relying solely on visually grounded knowledge is often insufficient for compositional reasoning. By combining heterogeneous sources, MI-RAG enables the model to perform compositional rea-

soning more effectively, leveraging diverse and relevant evidence to refine its search and generate more accurate reasoning records for the final answer.

# 6 ANALYSIS

## 6.1 SCALING RETRIEVERS

Building upon the established benefits of scaling MLLMs, we now investigate the impact of scaling the retriever. We investigated this by employing a larger vision encoder (SigLIP2-g) and a text embedding model (Qwen3-Embedding-0.6B). On the InfoSeek validation split, when using Gemini-2.5-Flash for MI-RAG, this enhanced retrieval improves accuracy to 66.7% (+6.4%) and cumulative recall to 95.2% (+3.1%). Similarly, on the Encyclopedic VQA test split, the scaled retriever paired with the Gemma-3-4B enhances accuracy to 50.2% (+1.9%) and cumulative recall to 65.7% (+12.1%). These findings confirm that improving retrieval capability provides significant benefits for MI-RAG's performance, validating the synergistic design.

## 6.2 ANALYSIS OF ITERATIVE PERFORMANCE

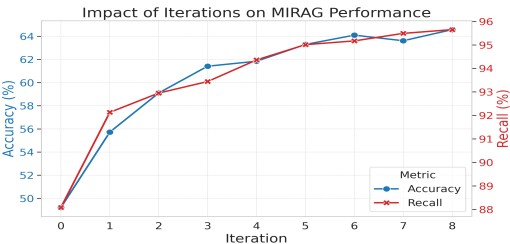

Figure 2: Accuracy and recall of MI-RAG on InfoSeek subset across 9 iterations.

We analyze how iterative refinement impacts MI-RAG's performance by measuring accuracy and recall across iterations. As shown in Figure 2, performance improves consistently with each step. The initial iterations deliver significant gains. Although the rate of improvement moderates in later steps, the model continues to achieve substantial gains in accuracy and recall. This performance improvement comes with an associated computational cost for each iteration, which we detail in Appendix F.

## 6.3 TOP-k SENSITIVITY ANALYSIS

Table 9: Top-k sensitivity analysis on the InfoSeek validation subset. $k$ indicates the number of retrieved image-text pairs.

| Model | Metric | $k = 10$ | $k = 20$ |
|---|---|---|---|
| Gemma-3-4B | Acc | 52.66 | 48.95 |
| | Recall | 90.82 | 94.52 |
| Gemini-2.5-Flash | Acc | 61.84 | 65.70 |
| | Recall | 94.36 | 95.97 |

Our sensitivity analysis on the InfoSeek validation subset (Table 9) reveals that the benefit of a larger retrieval budget is only beneficial if the MLLM is capable enough to handle the additional context. As shown in Table 9, the results reveal that simply increasing the retrieval budget is not always beneficial. For instance, Gemma-3-4B shows a degradation in accuracy when $k$ is increased from 10 to 20, suggesting that the additional context may potentially distract, hindering its ability to gather the correct evidence.

# 7 CONCLUSION

In this work, we introduce MI-RAG, a novel multimodal iterative RAG framework that addresses key challenges in knowledge-intensive VQA. MI-RAG employs a reasoning-guided multi-query transformation that dynamically explores multiple facets of an entity and its related textual knowledge across modalities. To facilitate compositional reasoning, its iterative process uses this retrieved evidence to progressively synthesize visual entities and corresponding textual knowledge, effectively integrating a diverse set of factual links. Our results demonstrate that MI-RAG is an effective and scalable paradigm for advancing compositional reasoning in knowledge-intensive VQA. Furthermore, our analysis indicates clear pathways for future work, such as scaling components by integrating web-scale search or dynamically adjusting the retrieval budget based on model capability.

## 8 ETHICS STATEMENT

Our work introduces new methods for multimodal retrieval-augmented generation for knowledge-intensive visual question answering, and we acknowledge the associated ethical implications. Potential risks include the propagation of biases from the underlying knowledge bases, which can lead to the generation of inaccurate or misleading information. To mitigate these concerns, our framework is intended for research purposes and requires careful evaluation before any deployment. Our approach uses only publicly available datasets and knowledge bases and does not involve private or personally identifiable information. We advocate for continued research on methods for bias detection and mitigation to ensure responsible development of these technologies.

## 9 REPRODUCIBILITY STATEMENT

To ensure full reproducibility, we provide the main algorithm code for MI-RAG in the supplementary materials. This includes comprehensive scripts for preparing the evaluation data, constructing the knowledge bases, and executing the main experiments. All implementation details, including our core algorithm, are described in Section 3 and detailed in Algorithm 1. The experimental setup, including datasets, knowledge bases, model variants, and evaluation metrics, is thoroughly outlined in Section 4.1.

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
