# OpenReview forum: "Multimodal Iterative RAG for Knowledge-Intensive Visual Question Answering"
_ICLR.cc/2026/Conference — ICLR 2026 Conference Withdrawn Submission_

### Official Review · Reviewer_pNo5 · 2025-10-18

**Soundness:** 2
**Presentation:** 1
**Contribution:** 2
**Rating:** 2
**Confidence:** 4

**Summary:**

The paper proposes MI‑RAG, an iterative retrieval‑augmented generation pipeline for knowledge‑intensive VQA. The system alternates between (i) reasoning‑guided multi‑query transformation (expanding and generating sub‑queries) and (ii) joint retrieval from two heterogeneous knowledge bases (a text‐only Wikipedia index and a multimodal KB of image–text pairs).

**Strengths:**

S1. Clear problem framing. The paper targets knowledge‑intensive VQA and correctly identifies failure modes of single‑pass multimodal RAG. Figures 1–2 and Algorithm 1 make the high‑level loop easy to follow.

S2. Simple, modular recipe. The method is mostly plug‑and‑play: prompt‑based query expansion/generation; lightweight retrievers (SigLIP2 image embeddings + ModernBERT/E5 text embeddings); and a generic VLM answerer.

S3. Comprehensive benchmarks. Evaluations cover Encyclopedic‑VQA, InfoSeek, and OK‑VQA with recall and accuracy metrics and include ablations over iterations, KB configuration, and retrieval budget.

**Weaknesses:**

W1. Low novelty: iterative RAG is not new.
The core idea—iteratively refine queries and interleave retrieval with reasoning—is well‑established in the text‑only literature (e.g., Self‑RAG and FLARE), and MI‑RAG largely ports this pattern to multimodal VQA with straightforward engineering (multi‑query + two KBs). There is no new learning objective, retriever architecture, or inference principle beyond prior iterative RAG/agent loops. The paper itself acknowledges the trend (§2.2). Recent exemplars include Self‑RAG (iterative retrieval, critique, and refinement) and FLARE (active, confidence‑triggered iterative retrieval). The paper neither positions itself clearly against these nor demonstrates a qualitatively different mechanism.

W2. Section 3 is too rough—thin technical contribution.
- Informal multi‑query transformation. §3.3 describes “query expansion” and “query generation” only as prompt templates; there is no algorithmic detail (e.g., uncertainty estimates, planning, or termination criteria) beyond repeating the loop in Algorithm 1.
- Ad‑hoc multimodal retrieval. §3.4 averages a text‑to‑text score with a fixed image‑to‑image similarity between the input and candidate image, “maintaining constant visual relevance across iterations.” This heuristic is under‑justified, ignores cross‑iteration visual disambiguation, and is not compared to stronger late‑interaction or cross‑modal reranking baselines.
- Reasoning records lack formalism. §3.5–§3.6 compress diverse evidence into free‑form summaries without any structure (e.g., entity‑relation graphs, uncertainty annotations), yet these records drive subsequent retrieval. The paper provides prompts (Appendix C) but no analysis of noise accumulation or error‑correction.

Overall, §3 enumerates components rather than delivering a technically rigorous method; most pieces are standard prompts or simple scoring rules.

W3. Missing/insufficient related‑work comparisons.
While §2 cites several multimodal RAG works, key iterative RAG baselines that the method emulates are absent from experiments and only weakly discussed:

- Self‑RAG (NeurIPS 2023): retrieval + critique + refinement loop; a canonical baseline for iterative RAG. https://arxiv.org/pdf/2310.11511
- FLARE / Active RAG (2023): confidence‑triggered iterative retrieval and query rewriting; a direct conceptual predecessor to MI‑RAG’s “reasoning‑guided multi‑query”. https://arxiv.org/pdf/2305.06983
- CRAG (Corrective RAG) (2024): evaluator‑guided corrective retrieval; relevant to the paper’s claim of mitigating irrelevant content but not discussed or compared. https://arxiv.org/pdf/2401.15884
- RAPTOR (2024): hierarchical retrieval via recursive summarization trees; highly relevant to MI‑RAG’s “compositional reasoning over diverse knowledge” but unmentioned. https://arxiv.org/pdf/2401.18059

W4. Experimental concerns.

W4.1 Fairness & apples‑to‑apples:
Several baselines are fine‑tuned (✓ in Tables 1–2) while MI‑RAG is “vanilla,” but differences in retrieval budgets and reranking remain uncontrolled (e.g., MI‑RAG retrieves up to 20 passages + 10 image–text pairs per iteration; Tables 1–2 report cumulative R@5 across iterations). Without normalizing budgets or reporting tokens/iteration by method, a recall lead may simply reflect more retrieval opportunities.

W4.2 Dependence on proprietary models:
Top accuracy numbers rely on Gemini‑2.5‑Flash and GPT‑4o. Reproducibility is limited and it’s unclear how much gain comes from the answerer’s capability vs. the proposed MI‑RAG loop (Table 3 and Table 4 show large gaps across answerers). A controlled study with only open models would make the claim stronger.

W4.3 Multimodal KB and potential leakage:
The multimodal KB contains 2M image–text pairs from Encyclopedic‑VQA (§4.1). Since the evaluation includes Encyclopedic‑VQA test, the paper must explicitly guarantee no leakage (e.g., no test images/near‑duplicates in the KB; no answer strings embedded in the paired text). This is not analyzed.

**Questions:**

See the above weaknesses W1-W4

---

### Official Review · Reviewer_wg47 · 2025-10-25

**Soundness:** 3
**Presentation:** 2
**Contribution:** 2
**Rating:** 4
**Confidence:** 4

**Summary:**

The paper addresses the common problem of “insufficient single-step retrieval” in knowledge-intensive Visual Question Answering (VQA) by proposing a Multimodal Iterative RAG (MI-RAG) framework. The core idea is that in each iteration, the model uses existing reasoning records to generate multiple queries (multi-query), performs joint retrieval across heterogeneous knowledge bases (textual KB and multimodal KB), and then integrates and records the newly retrieved knowledge to continuously refine the reasoning record. Finally, the model produces an answer based on the accumulated reasoning trajectory. The algorithmic process is presented in Algorithm 1, which includes key steps such as generating the initial description D, query expansion Q^E, sub-query generation Q^G, cross-KB joint retrieval, and iterative reasoning record updates. The authors evaluate the method on Encyclopedic VQA, InfoSeek, and OK-VQA, reporting improvements in recall and accuracy, and conduct ablation studies to verify the synergistic effects of “multi-query + heterogeneous KB + iteration.”

**Strengths:**

1) The paper clearly identifies the limitations of the single “retrieve-then-read” paradigm in knowledge-intensive VQA and transfers the idea of iterative RAG from the text domain to the multimodal setting, emphasizing the necessity of multi-query and cross-modal compositional reasoning. The motivation is sound but lacks strong originality.

2) The two key components of MI-RAG—reasoning-guided multi-query generation and joint retrieval across heterogeneous KBs—appropriately target the challenges of retrieval coverage/diversity and composable evidence. The overall algorithmic design is coherent, with detailed implementation aspects such as textual/multimodal similarity computation and retrieval quota allocation.

**Weaknesses:**

1) Several of the best results rely on large or commercial models (Gemini-2.5-Flash, GPT-4o) as answerers or for generating reasoning traces, while some baselines use open-source mid-sized models with additional fine-tuning or reranking.

2) The multimodal KB is constructed from 2 million image-text pairs in Encyclopedic VQA. If the test questions overlap significantly with KB pairs, the system may directly retrieve near-duplicate answers, which undermines interpretability and external validity. It is recommended to provide strict de-duplication or category filtering, along with supporting statistics, and also include results using a non-overlapping multimodal KB for comparison.

3) The paper demonstrates the iterative process with a fixed N=4, but the convergence criterion and adaptive mechanism for “early stopping or further iteration” under different levels of difficulty are not clearly defined. Although the appendix provides a cost analysis, it lacks latency or computational cost statistics per question.

4) While the limitations of EM are discussed and BEM/CEM are introduced, the impact of answer style (long explanations vs. short entities) on evaluation metrics, as well as the error distribution across different question types (single-hop/multi-hop, entity/attribute), is not sufficiently analyzed. It is suggested to include per-type curves and confidence intervals.

5) When using Pseudo-Relevance Recall (PRR) for retrieval evaluation, it is recommended to include manual sample verification or correlation analysis with entity-level recall.

6) The multimodal retrieval similarity is computed as the average of image-image (fixed) and text-text similarity. Although simple and effective, this lacks systematic comparison with strong rerankers or interaction-based matching methods. Furthermore, it does not discuss how false positives (e.g., visually similar but semantically different items) may interfere with the reasoning record. It is suggested to add cross-modal interaction scoring or a learned reranker as a baseline.

**Questions:**

1. What kind of hash-based or perceptual de-duplication was applied between the multimodal KB and the test set? Could the authors provide statistics on image/entity overlap rates and results under a “non-overlapping KB” scenario? (This is crucial for external validity.)

2. Beyond using a fixed N, can early stopping be determined based on marginal information gain or uncertainty? Have the authors tried strategies linking iteration depth to question difficulty or retrieval diversity, and what were the outcomes?

3. Fair comparison: Can the authors provide a fully open-source, scale-matched (e.g., 7B-parameter answerer, same retriever) end-to-end comparison with recent strong baselines, including significance testing?

4. Error analysis: Can the authors present fine-grained statistics of CEM/EM/BEM by question or answer type, along with representative failure cases, and indicate the marginal contribution of each iteration to the final result?

5. Other issues mentioned in the Weaknesses section also require clarification.

**Details Of Ethics Concerns:**

Not required.

---

### Official Review · Reviewer_Ayzh · 2025-10-31

**Soundness:** 2
**Presentation:** 3
**Contribution:** 2
**Rating:** 4
**Confidence:** 4

**Summary:**

This paper proposes MI-RAG, a multimodal iterative retrieval-augmented generation framework for knowledge-intensive visual question answering. MI-RAG employs an iterative loop where it dynamically generates multiple queries guided by its reasoning state to retrieve diverse knowledge across heterogeneous sources. This retrieved knowledge is then synthesized to progressively refine its understanding.

**Strengths:**

- The proposed method demonstrates strong performance gains across multiple VQA benchmarks.
- The core focus on multi-iteration retrieval is both timely and meaningful, addressing a recognized limitation in single-pass retrieval-augmented systems.

**Weaknesses:**

1. Heavy Reliance on Prompt Engineering: The framework relies heavily on prompt engineering for key components such as query transformation and knowledge synthesis. Critical components—including query transformation, knowledge fusion across heterogeneous sources, and multi-step reasoning—are implemented through prompt design without underlying architectural innovations or principled methodologies. While effective, this approach may limit the generalizability and interpretability of the method. It would be valuable to see a more systematic analysis of why certain prompts work, or to propose design principles that could guide future work.
2. Lack of In-depth Analysis on Iteration Mechanics: While iterative refinement is central to the method, the paper lacks a mechanistic analysis of how each iteration contributes to final performance. The paper would benefit from a deeper analysis of the iterative process itself—for example, how the reasoning records evolve across iterations, how the model integrates new information, and whether certain types of questions benefit more from multiple iterations.
3. Absence of Reusable Methodological Insights: The empirical nature of prompt-centric design makes it difficult to extract reusable principles or transferable methodologies. In its current form, the work reads more like an engineering report than a research paper offering generalizable insights for the community.

**Questions:**

1. How does the framework handle cases of erroneous or conflicting retrievals? Does the MLLM exhibit robustness in synthesizing contradictory evidence, and are there mechanisms to detect or mitigate such scenarios?
2. What ensures diversity in retrieval across iterations? Are there explicit constraints or prompt designs to prevent the model from repeatedly retrieving similar or redundant information in subsequent rounds?

---

### Official Review · Reviewer_trvQ · 2025-11-01

**Soundness:** 1
**Presentation:** 2
**Contribution:** 2
**Rating:** 2
**Confidence:** 3

**Summary:**

This paper introduces MI-RAG, a Multimodal Iterative RAG framework designed to improve knowledge-intensive visual question answering (VQA). The authors identify the limitation of conventional single-pass RAG systems, which often fail to collect sufficient or relevant external knowledge in one shot. To address this, MI-RAG iteratively refines both the retrieval and reasoning processes. At each iteration, the model formulates reasoning-guided multi-queries to explore multiple facets of knowledge, retrieves from heterogeneous sources, and synthesizes the retrieved information into a progressively enriched reasoning record. The proposed system is evaluated on several multimodal VQA benchmarks (Encyclopedic VQA, InfoSeek, OK-VQA), achieving superior retrieval recall and answer accuracy compared to baselines.

**Strengths:**

1. The proposed iterative reasoning–retrieval loop is well-formulated, showing a logical progression from query formulation to knowledge synthesis. This design aligns well with human-like reasoning processes.
2. The paper is clearly written.

**Weaknesses:**

1. Given the large body of prior work exploring retrieval performance in RAG systems, the novelty of MI-RAG is somewhat incremental. I'd like to view the approach as one of many recent RAG variants focusing on iterative improvement, which weakens its originality.

2. While the paper compares to standard multimodal baselines, it does not include strong RAG-based retrieval–reasoning approaches. Without these comparisons, it is difficult to isolate the improvement from the proposed iterative refinement mechanism.

3. Evaluation remains largely tied to VQA datasets. Extending to other multimodal RAG tasks would better demonstrate scalability.

**Questions:**

NA

---

### Note · Authors · 2025-11-19

**Comment:**

I have read and agree with the venue's withdrawal policy on behalf of myself and my co-authors.

**Withdrawal Confirmation:**

I have read and agree with the venue's withdrawal policy on behalf of myself and my co-authors.